Using transcriptome sequencing (RNA-Seq) to screen genes involved in β-glucan biosynthesis and accumulation during oat seed development

Qi Bing jie qbjzzy@126.com
Ji Ming xue
He Zhu qing
College of Agriculture, Inner Mongolia Agricultural University , Hohhot , Inner Mongolia , China
Kutlu Imren
Electronic publication date: 2024 Sep 25
Publication date: 2024
Volume: 12
Electronic Location ID: e17804
Received 2023 Nov 9; Accepted 2024 Jul 3
Copyright: ©2024 Qi et al.
Copyright year: 2024
Copyright holder: Qi et al.
License: This is an open access article distributed under the terms of the Creative Commons Attribution License, which permits unrestricted use, distribution, reproduction and adaptation in any medium and for any purpose provided that it is properly attributed. For attribution, the original author(s), title, publication source (PeerJ) and either DOI or URL of the article must be cited.
License URL: https://creativecommons.org/licenses/by/4.0/

Keywords: Oat, β-glucan, Biosynthesis, Accumulation, Transcriptome sequencing

Funding: The Inner Mongolia Autonomous Region Science and Technology Plan Project NO.2020GG0037 This work was supported by the Inner Mongolia Autonomous Region Science and Technology Plan Project (NO.2020GG0037). The funders had no role in study design, data collection and analysis, decision to publish, or preparation of the manuscript.

==============================
Oat (Avena sativa L.) is an annual grass that has a high nutritional value and therapeutic benefits. β-glucan is one of the most important nutrients in oats. In this study, we investigated two oat varieties with significant differences in β-glucan content (high β-glucan oat varieties BY and low β-glucan content oat variety DY) during different filling stages. We also studied the transcriptome sequencing of seeds at different filling stages. β-glucan accumulation was highest at days 6-16 in the filling stage. Differentially expressed genes (DEGs) were selected from the dataset of transcriptome sequencing. Among them, three metabolic pathways were closely related to the biosynthesis of β-glucan by Gene Ontology (GO) and Kyoto Encyclopedia of Genes and Genomes (KEGG) analysis, including xyloglucan:xyloglucosyl transferase activity, starch and sucrose metabolism, and photosynthesis. By analyzing the expression patterns of DEGs, we identified one CslF2 gene and 32 transcription factors. Five modules were thought to be positively correlated with β-glucan accumulation by weighted gene co-expression network analysis (WGCNA). Moreover, the expression levels of candidate genes obtained from the transcriptome sequencing were further validated by quantitative real-time PCR (RT-qPCR) analysis. Our study provides a novel way to identify the regulatory mechanism of β-glucan synthesis and accumulation in oat seeds and offers a possible pathway for the genetic engineering of oat breeding for higher-quality seeds.

Introduction

Oats (Avena sativa L.) are a high-quality whole grain and one of the world’s most important crops that may have certain health benefits (Ahmad, Dar & Habib, 2014; Wu et al., 2017). A large part of the health benefits of oats comes from their richness in the soluble dietary fiber, β-glucan; oats have the second highest content of β-glucan in cereals after barley (Chen et al., 2016). β-glucan is a natural water-soluble polysaccharide unique to the grass family, which is a collective name for polymeric glucose polymers made from D-glucopyranose residues linked by β- (1-3) and β- (1-4) glycosidic bonds (Wolever et al., 2010). Numerous studies have confirmed that β-glucan has a high utilization value in various fields. As early as 1997 the US Food and Drug Administration (FDA) approved a health claim highlighting that β-glucan in oats may reduce the risk of coronary heart disease (https://www.federalregister.gov/documents/1997/01/23/97-1598/food-labeling-health-claims-oats-and-coronary-heart-disease). Mathews, Kamil & Chu (2020) reviewed and analyzed 49 clinical trials published from 1997 to February 2019; they found that most of the trials indicated that oat β-glucan was effective in lowering total cholesterol, and these results support the FDA’s health claims. In addition to this, β-glucan may prevent diabetes. β-glucan has a natural reticular structure and is easily wrapped by proteins and starch to form a complex structural matrix which reduces enzyme accessibility and leads to reduced starch metabolism, thus lowering blood glucose (Zhang, Luo & Zhang, 2017). β-glucan also has a high viscosity and may act as a physical barrier to the body’s ability to absorb glucose (Abbasi et al., 2016). The high viscosity of β-glucan may also act as a mechanism for its hypotensive effect (Bai et al., 2019). Several studies have shown that β-glucan plays an important role in promoting apoptosis and anti-tumor properties (Mo et al., 2017; Choromanska et al., 2015; Parzonko et al., 2015).

Many researchers have begun to explore genes that may regulate the synthesis and accumulation of β-glucan in oats. However, the mechanism of oat β-glucan synthesis and metabolism has not been fully explained. The CSlH gene (HvCSlH) in barley was identified and introduced into Arabidopsis by genetic transformation, resulting in the biosynthesis of β-glucan and further deposition on the cell wall in Arabidopsis. HvCSlH may be the major gene controlling the biosynthesis of β-glucan (Doblin et al., 2009). CslF 6 has been proven to be the major gene to regulates the biosynthesis of β-glucan in barley seeds (Burton et al., 2011; Goudar et al., 2020). This gene has also been identified in wheat and has been shown to affect wheat endosperm β-glucan content (Nemeth et al., 2010). However, the key enzyme genes that determine the metabolism of β-glucan in oats are still unknown.

In recent years, most of the research on oats has focused on the mining of antagonistic genes (Wu et al., 2018; Zhang et al., 2021a). In contrast, there have been few studies on genes related to β-glucan biosynthesis and accumulation during oat seed development. As a β-glucan-rich health food, oats can well meet the current market demand for a healthy diet. The breeding goal of oats should be more responsive to the market demand. Therefore, the genes related to β-glucan biosynthesis and accumulation in oat seeds should be explored and the expression patterns of related genes should be analyzed.

In this study, we analyzed the transcriptome during oat seed grain development to explore the regulatory mechanisms of β-glucan synthesis in oat seed grain. Our study will provide new ideas for the key regulatory mechanisms of β-glucan synthesis and accumulation in oat seed grain. Meanwhile, it will provide a theoretical basis for further breeding of high β-glucan oat varieties.

Materials & Methods

Plant materials and sampling of oat seeds

The plant material used in this study was the high β-glucan content variety BY and the low β-glucan content variety DY. These varieties were selected from among 130 oat germplasm resources (Table S1). Seeds were planted in the experimental field of Inner Mongolia Agricultural University, Hohhot, Inner Mongolia (40°80′77″N, 111°62′29″E). The two varieties were sowed on April 20 and April 30, 2021, respectively, to achieve a consistent flowering period. A randomized zonal design with three replications was used. The area of the plot is 5 m*5 m = 25 m2 with a row spacing of 30 cm. A total of 50 plants of each variety with the same growth were randomly selected at the spike stage for numbering and listing. To ensure the consistency of the seed growth period, the other whorl layers were cut off with scissors and only the first whorl layer was retained. We treated day 7 after flowering as day 1 of the seed-filling period (1 d). Seed samples were collected at five seed-filling periods: 1 d (BY1 and DY1), 6 d (BY2 and DY2), 11 d (BY3 and DY3), 16 d (BY4 and DY4), and 21 d (BY5 and DY5), and harvest mature seeds (26 d). Seeds collected on 1 d were used as the control. The samples were placed into a sterile frozen storage tube, quickly frozen with liquid nitrogen, and returned to the laboratory for storage in an ultra-low temperature refrigerator (−80 °C). Three biological replicates were taken for each sample, and a total of 30 samples were obtained.

Determination of β-glucan content in oat seeds

The β-glucan content of oat seeds was determined using the Megazyme® mixed-linkage β-glucan assay kit (Biostest, Beijing, China), and the experiment was performed according to the kit instructions with three replicates.

RNA extraction and cDNA library construction

The total RNA of the oat seeds was extracted using RNA Prep Pure Plant Kit (Tiangen, Tianjin, China), using approximately 0.1 g of seeds. RNA integrity was determined by 1% agarose gel electrophoresis, and RNA integrity values (>6.5) were detected by Agilent 5400 Bioanalyzer. RNA concentration and purity were measured using a NanoDrop 2000 spectrophotometer (Thermo Fisher Scientific, Waltham, MA, USA). Oligo(dT) magnetic beads were used to enrich the mRNA with poly-A tails, and mRNA was chemically cleaved using divalent metal ion solutions. Randomly fragmented mRNA was used as a template to synthesize the first and second cDNA strands. cDNA fragments were enriched with 14 PCR cycles and library concentrations were precisely quantified to ensure that the libraries were available for subsequent sequencing. A total of 30 cDNA libraries were constructed.

RNA-seq data processing

High-throughput double-end sequencing was performed using Illumina NovaSeq 6000 (Illumina, USA). Reads with connectors, undeterminable base information, low quality (Q<20), and lengths less than 20 bp were filtered, and the data were assembled and spliced for subsequent bioinformatics analysis. The obtained clean data were compared to the reference genome (https://wheat.pw.usda.gov/GG3/graingenes-downloads/pepsico-oat-ot3098-v2-files-2021) using HISAT2 (Mortazavi et al., 2008). Fragments per kilobase of transcript per million mapped reads (FPKM) were calculated to evaluate the gene expression level of each sample. Differentially expressed gene (DEGs) analysis was performed between comparative combinations using DESeq2 (1.20.0) software. P-adj ≤0.05 and —log2FC—≥1 were set as the thresholds of significant differential expression (Mao et al., 2005). Gene Ontology (GO) enrichment analysis, Encyclopedia of Genes and Genomes (KEGG) pathway enrichment analysis, and gene expression analysis were performed using the R package embedded in Novomagic (https://magic.novogene.com/). The data were plotted graphically.

Weighted gene co-expression network analysis

To identify DEGs associated with β-glucan more comprehensively, weighted gene co-expression network analysis (WGCNA) was applied to RNA-seq data. The expression levels of differentially expressed genes in different samples were clustered by WGCNA, and correlation analysis was performed to find the set of genes directly associated with β-glucan content in oat seeds.

Detecting the gene expression level by quantitative real-time PCR (RT-qPCR) analysis

Total RNA was extracted from oat seeds using RNA prep Pure Plant Kit (Tiangen, Tianjin, China) using approximately 0.1 g of seeds. RNA integrity was determined by 1% agarose gel electrophoresis, and RNA concentration and purity were determined by NanoDrop2000 spectrophotometer. RNA with an A260/280 ratio of 1.8–2.0 and an A260/A230 ratio of 2–2.5 may be used for subsequent experiments (Table S2). The actin (KP257585.1) gene of oats was used as the internal reference gene, and primers were designed for the reference genes and differentially expressed genes according to the RT-qPCR primer design principle using Primer 3 Plus (https://www.primer3plus.com/), which were synthesized by Sangon Biotech (Shanghai, China), and the specificity was verified in pre-experiments (Table S3). The first strand of cDNA was synthesized with TIANGEN® FasKting cDNA (Tiangen, Tianjin, China) First Strand Synthesis Kit (RNA 2 µl; 5 ×gDNA Buffer 2 µl; RNase-Free ddH2O 6 µl; 10 ×King RT Buffer 2 µl; FastKing RT Enzyme Mix 1 µl; FQ-RT Primer Mix 2 µl; RNase-Free ddH2O 5 µl.), incubated at 42 °C for 15 min, then 95 °C for 3 min. The cDNA obtained was subjected to fluorescence quantitative PCR using the SGExcel FastSYBR qPCR Mix (Sangon Biotech, Shanghai, China) KitI (1 ×MonAmpTM SYBR® Green qPCR Mix 10 µl; 10 µM forward primer 0.4 µl; 10 µM reverse primer 0.4 µl; cDNA 2 µl; RNase-Free Water 7.2 µl). The reaction procedure was: 95 °C pre-denaturation for 30 s, denaturation at 95 °C for 30 s, annealing at 60 °C for 40 s, extension at 72 °C for 30 s. The last three steps were performed for a total of 40 cycles, three replicates for each sample. The lysis curves and amplification curves were collected to verify primer specificity. Finally, the relative expression values were calculated by the 2−ΔΔCq method. To ensure data availability, all 30 samples were subjected to four concurrent identical technical replications.

Statistical analysis

We analyzed the data using one-way ANOVA for significance using IBM SPSS Statistics 25 (Meyers, Gamst & Guarino, 2013). Origin 2021 (https://www.originlab.com/) was used by us to plot dot lines and bar plots. We used Novomagic (https://magic.novogene.com/) to analyze the transcriptome data and plot all charts.

Results

Dynamics of β-glucan content in oat seeds at different filling stages

In order to study the accumulation pattern of β-glucan in oat seeds during the filling period (1 d–26 d), we determined the β-glucan content in BY and DY seeds at different filling periods (Fig. 1, Table S4). The results showed that the β-glucan content of the two varieties maintained a gradually increasing trend during the filling period. The accumulation rate of β-glucan in both varieties showed the characteristic of “fast-slow-fast”. The period from days 6–16 was the stage with the fastest accumulation rate of β-glucan, and more than 72% and 70% of the β-glucan in BY and DY, respectively, were accumulated in this stage. From 6 d onwards, both varieties showed significant differences (P ≤ 0.001) in β-glucan content. The β-glucan content of BY was 5.414% in mature seeds (26 d), which was significantly higher than that of DY (2.092%).

Figure 1 Dynamic change of β-glucan content during grain filling of oat.

Error bars indicates the standard deviation. Significant difference between varieties at p ≤ 0.05 were indicated by different letters.

Transcriptome sequencing and data quality control

Illumina sequencing data quality analysis is shown in Table 1. There were 2,216,337,008 original downstream sequences obtained from 30 samples, with a total valid sequence data volume of 324.09 G and data volume variability between 10.02 G–11.66 G; the data error rate of all 30 samples was about 0.03%; the Q30 content was between 92.73%–93.73; the GC content ranged from 53.14%–57.66%. The sequencing was of high quality and fully satisfied the requirements of subsequent analysis.

Table 1 Sequencing statistics and quality control.

Sample	Raw reads	Clean reads	Clean bases	Error rate(%)	Q20 (%)	Q30 (%)	GC pct (%)	
BY1_1	7,4185,008	7,2233,220	10.83G	0.03	97.35	92.9	53.3	
BY1_2	7,1631,146	6,9785,896	10.47G	0.03	97.27	92.74	54.12	
BY1_3	6,9908,164	6,8592,802	10.29G	0.03	97.37	92.98	54.57	
BY2_1	7,0013,118	6,8479,510	10.27G	0.03	97.28	92.75	53.69	
BY2_2	7,0135,848	6,8560,432	10.28G	0.03	97.43	93.05	54.06	
BY2_3	7,8781,784	7,6968,516	11.55G	0.03	97.67	93.55	53.77	
BY3_1	6,9458,960	6,7563,402	10.13G	0.03	97.39	92.92	53.46	
BY3_2	7,6028,496	7,4394,622	11.16G	0.03	97.39	92.89	53.21	
BY3_3	7,7949,008	7,6103,246	11.42G	0.03	97.28	92.73	54.16	
BY4_1	8,0734,234	7,8674,852	11.8G	0.03	97.42	93.08	55.04	
BY4_2	7,8190,806	7,6274,062	11.44G	0.03	97.74	93.73	54.11	
BY4_3	7,1774,370	7,0299,766	10.54G	0.03	97.47	93.11	54.33	
BY5_1	7,0329,084	6,8642,898	10.3G	0.03	97.62	93.51	57.66	
BY5_2	7,6884,354	7,5647,058	11.35G	0.03	97.77	93.77	56.14	
BY5_3	7,5646,140	7,4017,414	11.1G	0.03	97.69	93.6	55.65	
DY1_1	7,5644,434	7,3267,134	10.99G	0.03	97.38	93.01	53.14	
DY1_2	7,5149,778	7,2900,980	10.94G	0.03	97.43	93.12	54.82	
DY1_3	7,2337,450	7,0340,732	10.55G	0.03	97.29	92.86	55.08	
DY2_1	7,0269,126	6,8432,396	10.26G	0.03	97.52	93.23	53.72	
DY2_2	6,8949,650	6,7211,240	10.08G	0.03	97.48	93.17	54.14	
DY2_3	7,5718,542	7,4084,990	11.11G	0.03	97.67	93.63	54.58	
DY3_1	6,8661,546	6,6813,470	10.02G	0.03	97.4	93.05	54.33	
DY3_2	6,9799,034	6,7887,538	10.18G	0.03	97.52	93.27	53.44	
DY3_3	7,6960,750	7,4613,082	11.19G	0.03	97.41	93.11	54.6	
DY4_1	8,0227,042	7,7760,732	11.66G	0.03	97.61	93.41	55.06	
DY4_2	7,5592,742	7,3050,936	10.96G	0.03	97.5	93.22	54.26	
DY4_3	7,4070,330	7,2311,160	10.85G	0.03	97.57	93.33	54	
DY5_1	7,0295,922	6,8510,546	10.28G	0.03	97.46	93.22	57.54	
DY5_2	7,2055,370	7,0244,784	10.54G	0.03	97.53	93.29	54.82	
DY5_3	7,8954,772	7,6990,606	11.55G	0.03	97.44	93.11	54.6	

To assess the correlation between the three biological replicates, we performed Pearson linear correlation analysis on the sequencing data of three biological replicates of each variety and plotted the correlation heat map (Fig. 2A). The results indicated that the gene expression similarity among the three biological replicates of the 10 samples was high enough for subsequent analysis of differential genes. Principal component analysis was then performed (Fig. 2B). The tight clustering between the three biological replicates of each sample indicated that the samples were reproducible, and the two varieties were separated from each other, indicating that there were significant differences in transcriptional programs between the two varieties at each filling stage.

Figure 2 Dynamic change of β-glucan content during grain filling of oat.

(A) Pearson correlation coefficient analysis of gene expression level in two oat varieties at different filling stages. (B) Principal component analysis of gene expression levels in two oat varieties at different grain filling stages.

Screening and functional annotation of DEGs among different species

A two-by-two comparison of BY and DY was performed for each filling period to investigate the transcriptional regulatory mechanisms associated with the synthesis and accumulation of β-glucan in both genotypes. A total of 8,443, 8,866, 7,137, 8,875, and 5,522 DEGs were obtained for the five comparison groups (BY1 vs. DY1, BY2 vs. DY2, BY3 vs. DY3, BY4 vs. DY4 and BY5 vs. DY5, respectively), of which 2,066 were common to the five periods (Fig. 3A, Table S5). Further analysis of the DEGs revealed that the number of up-regulated genes was greater than the number of down-regulated genes on 1 d, 6 d, and 21 d of grouting; this trend was reversed on 16 d, and the number of up-regulated and down-regulated genes was almost identical on 11 d (Fig. 3B). A cross-sectional comparison between the five comparison groups revealed that 1,459 DEGs were consistently up-regulated as the grouting process proceeded and 605 genes were consistently down-regulated with the perfusion process (Table S6).

Figure 3 Screening and enrichment of DEGs among different varieties.

(A) Venn diagram of DEGs at five filling stages among different varieties. (B) Histogram of DEGs at five filling stages among different varieties. (C) KEGG Pathway dot diagram with significant enrichment of DEGs in five filling stages among different varieties.

All of the screened DEGs were enriched for GO functions, and a total of 213 GO-terms were significantly (P-value < 0.05) enriched for the five comparison groups (Fig. S1). These terms involved the glucan metabolic process, cellular polysaccharide metabolic process, cellular glucan metabolic process, cell wall, protein heterodimerization activity, xyloglucan:xyloglucosyl transferase activity, and nutrient reservoir activity GO.

KEGG pathway enrichment was performed for the DEGs screened in each comparison group and a total of 21 significantly (P-value < 0.05) enriched pathways were obtained (Fig. 3C), of which protein processing in the endoplasmic reticulum, pentose phosphate pathway and flavonoid biosynthesis were significantly enriched in at least two perfusion periods.

Screening and functional annotation of DEGs during seed development

To investigate the DEGs associated with the synthesis and accumulation of β-glucan in oat seeds at different filling stages, we used the gene expression levels of DY and BY at the 1 d of filling as controls. We compared the gene expression changes of the two varieties at 6, 11, 16, and 21 days of filling. A total of 77,534 DEGs were obtained from the eight comparison groups (BY2 vs. BY1, BY3 vs. BY1, BY4 vs. BY1, BY5 vs. BY1, DY2 vs. DY1, DY3 vs. DY1, DY4 vs. DY1, DY5 vs. DY1), of which 38,675 were obtained for BY and 38,859 for DY. A total of 1,557 and 1,656 DEGs were co-expressed for BY and DY at the four perfusion stages, respectively (Fig. 4A, Table S7). Further analysis of the expression of all DEGs revealed that the number of DEGs in both varieties tended to increase as the filling process proceeded, and the number of up-regulated genes was greater than that of down-regulated genes in each period (Fig. 4B, Table S8).

Figure 4 Screening and enrichment of DEGs during BY and DY grain development.

(A) Venn diagram of DEGs during BY and DY grain development. (B) Histograms of DEGs during BY and DY grain development. (C) KEGG pathway dot diagram with significant enrichment of DEGs during BY and DY grain development.

All screened DEGs were enriched for GO function, and DEGs of BY and DY were significantly (P-value < 0.05) enriched for 203 and 284 GO-terms, respectively (Fig. S2). The main GO entries included responses to abiotic stimulus, cell wall, and protein heterodimerization activity. Notably, DEGs of DY at 11 d, 16 d, and 21 d were significantly enriched for the GO term xyloglucan:xyloglucosyl transferase activity, whereas BY was significantly enriched in this entry only at 6 d.

KEGG pathway enrichment was performed for DEGs screened in each comparison group, and DEGs from BY and DY were significantly (P-value < 0.05) enriched into 16 and 18 pathways, respectively (Fig. 4C). Of these 34 significantly enriched pathways, 11 are common to both species, including starch and sucrose metabolism, pentose and glucuronate interconversions, protein processing in the endoplasmic reticulum, and galactose metabolism, protein processing in the endoplasmic reticulum, and galactose metabolism.

Candidate genes related to β-glucan biosynthesis and accumulation during seed development

Starch and sucrose metabolism (map00500) has been shown to be closely associated with β-glucan biosynthesis and accumulation, but β-glucan biosynthesis and accumulation is a complex biological process. We analyzed the expression pattern of each gene in the starch and sucrose metabolism pathway (Fig. 5A) to further explore the key genes for β-glucan biosynthesis and accumulation in oat seeds. The results showed that the expression of TRINITY_DN34723, TRINITY_DN15420, and TRINITY_DN34252, which encode glucan endonucleases, decreased in both genotypes of oats as the seeds developed, and the expression in BY was lower than that in DY, which was consistent with the change in β-glucan content of the seeds.

Figure 5 Heat map of DEGs associated with β-glucan biosynthesis and accumulation during BY and DY seed development.

(A) Heat map of the three glucan endonuclease genes in map00500. (B) Heat map of the four CslF genes.

It is well documented that members of two cellulose synthase-like (CslF and CslH) subfamilies are responsible for β-glucan synthesis (Liepman & Cavalier, 2012; Doblin et al., 2009; Schreiber et al., 2014). Therefore, we screened the DEGs of the five filling periods of BY and DY and identified a total of 56 CslF and seven CslH genes. However, only one CslF2, two CslF8, and one CslF9 gene reached a certain expression abundance (Fig. 5B). Among them, the gene TRINITY_DN16295, encoding CslF2, was expressed on average 4.6 times more than DY in all periods of BY, and the expression of this gene showed a continuous increase with the development of the seed filling process.

Transcription factors related to β-glucan biosynthesis and accumulation during seed development

Different transcription factors (TFs) regulate a range of life processes during the growth and development of a plant. To identify the TFs involved in regulating β-glucan biosynthesis and accumulation during oat seed development, we investigated the genes encoding TFs in the RNA-Seq dataset. The low β-glucan genotype oat DY was used as a control to compare the FPKM values of TFs at each developmental period. The results showed that there were a total of 2,256 up-regulated TFs expressed in the five comparison groups, of which 213 TFs were common to the five comparison groups (Fig. S3). These TFs belong to 41 families such as bHLH, WRKY, NAC, MYB-related, bZIP, and M-type-MADS, respectively (Table S9). We found that these 13 TFs belonged to 10 families including B3, FAR1 and NAC. Some of them were up-regulated expression in BY, while they were not expressed in DY. The expression pattern of another part in DY was consistent with that of BY, but the expression amount was much lower than that of BY (Fig. 6A). In addition to this, we found 19 TFs belonging to 13 families respectively whose expression first increased and then decreased (Fig. 6B). Their expression peaked at 11–16 d, which was consistent with the trend in the accumulation rate of β-glucan. Notably, five of these 19 TFs belonged to the Dof family in common.

Figure 6 Heat map of TFs associated with β-glucan synthesis and accumulation during BY and DY seed development.

(A) Heat map of TFs up-regulated with seed filling. (B) Heat map of TFs adjusted first upwards and then downwards with seed filling.

Weighted gene co-expression network analysis of DEGs

A total of 39,211 DEGs were clustered based on gene co-expression patterns. To make the constructed network more consistent with the scale-free topology, we chose five as the optimal soft threshold (Fig. S4), identified a network with 32 co-expression modules (Fig. 7A), and analyzed the characteristic genes of each co-expression module and their relationship with β-glucan content in oat seeds (Fig. 7B). Five modules associated with β-glucan were clearly identified (p < 0.05), namely the blue module (r = 0.93, p = 9 ×10−5), magenta module (r = 0.82, p = 0.004), light cyan module (r = 0.74, p = 0.01), white module (r = 0.70, p = 0.03) and dark green module (r = 0.67, p = 0.03). These modules were significantly and positively correlated with β-glucan content in oat seeds and contained 6,559, 1,223, 364, 80, and 139 DEGs, respectively.

Figure 7 WGCNA co-expression network and module-trait correlation analysis.

(A) DEGs module hierarchical clustering tree. (B) Correlations of the content of β-glucan in oat grains with WGCNA modules.

RT-qPCR validation of RNA-Seq data

We validated RT-qPCR on seven randomly selected genes using actin as an internal reference and performed a linear fit of RT-qPCR to RNA-Seq data, which showed that R2 was 0.81 and 0.48 in BY and DY, respectively. The Pearson correlation coefficients were 0.90 and 0.69, respectively. This indicates that the RNA-Seq results were reliable (Fig. 8, Table S10).

Figure 8 Correlation between qRT-PCR and RNA-seq data for DEGs related to β-glucan synthesis and accumulation in BY and DY at different stages of grain filling.

Error bars indicates the standard deviation.

Discussion

Oats are a nutritious whole grain. They are rich in high-quality carbohydrates and proteins with a balance of amino acids and contain many trace elements such as Se and Mn (Chen et al., 2021; Gulvady, Brown & Bell, 2013; Stewart & McDougall, 2014). In recent years, the β-glucan in cereal grains has gained attention for its potential health properties. However, the content of β-glucan in oat kernels under normal growing conditions is only about 35.92 g kg−1, which cannot satisfy the market demand for β-glucan (Bai et al., 2021). Therefore, it is necessary to breed new oat varieties with higher seed β-glucan content. In this study, we compared the transcriptome profiles of two oat genotypes with large differences in seed β-glucan content using RNA-Seq analysis to screen out some genes that may be related to β-glucan synthesis and accumulation in oat seeds, laying the foundation for revealing the molecular mechanism of β-glucan synthesis and accumulation in oat seeds.

In this study, we determined the β-glucan content of oat seeds of two different genotypes and found that there were large differences in the β-glucan content and accumulation rate between the two varieties. In general, the β-glucan content showed a gradual increase throughout the filling period. BY was greater than DY and the accumulation rate was highest at 6–16 d of filling and then slowed down. This may be because the process of oat grain filling is usually accompanied by the aging process of the leaves. The leaves are the main site of oat photosynthesis, and most of the nutrients of the seeds in the filling period depend on the assimilates produced by the photosynthesis of the leaves. Therefore, as the leaves age, photosynthesis is weakened and a series of vital reactions are also weakened, reducing the nutrient supply and slowing the filling rate of the seeds. The rate of seed filling becomes slower, and the endosperm cells do not increase after the milk ripening stage, thus causing the rate of β-glucan accumulation to become slower. Therefore, we consider the 6-16 d period for filling as a critical period for β-glucan accumulation in oat seeds.

It has been shown that the barley xyloglucan:xyloglucan transferase HvXET5 can catalyze the formation of covalent bonds between xyloglucan and cellulose substrates and between xyloglucan and (1,3;1,4)- β-D-glucan in vitro (Hrmova et al., 2007). The number of down-regulated genes that were enriched to the xyloglucan:xyloglucosyl transferase activity in this study was much larger than the up-regulated genes. During the seed filling process, more down-regulated genes were significantly enriched to the low β-glucan variety DY compared to the high β-glucan variety BY in this xyloglucan:xyloglucosyl transferase activity term. In a two-by-two comparison between BY and DY at each period, the genes that were enriched to this term gradually decreased over time. Therefore, we suggest that the genes enriched in this pathway play a negative regulatory role in β-glucan accumulation.

A regulatory relationship between carbohydrates and β-glucans has revealed a degree of association between starch and the biosynthetic pathway of β-glucan (Marcotuli et al., 2016; Munck et al., 2004). KEGG pathway analysis produced DEGs that were significantly enriched in starch and sucrose metabolism, galactose metabolism, photosynthesis, and pentose and glucuronide interconversion pathways. We screened three encoding genes (TRINITY_DN34723, TRINITY_DN15420, and TRINITY_DN34252) that encode glucan endonucleases in starch and sucrose metabolism (map00500). Their expression in both genotypes of oats decreased with seed development and was lower in BY than in DY, which was opposite to the trend of β-glucan content in seeds. Therefore, we believe that it is the high expression of these three genes encoding glucan endonucleases that negatively affects β-glucan accumulation in DY, resulting in lower β-glucan content in DY than in BY. No other functional genes related to β-glucan biosynthesis were found in this pathway. Geng et al. (2022) found that barley varieties with up-regulated photosynthesis-related genes also had elevated β-glucan content, thus concluding that photosynthesis was positively regulated by β-glucan accumulation in barley seeds. Zhang et al. (2021b) also found up-regulated of the glucan synthase AsCslF6 gene at high light intensities, demonstrating that β-glucan synthesis in oat leaves is positively regulated by light. However, with the accumulation of β-glucan in BY in this study, the majority of DEGs enriched in the photosynthetic pathway were down-regulated, probably due to the process of leaf senescence that often accompanies late grain filling and the consequent weakening of photosynthesis. Galactose metabolism was significantly enriched to a large number of DEGs in several periods, and the number of up-regulated genes was greater than the number of down-regulated genes, so we can reasonably speculate that there is a positive correlation between galactose metabolism and β-glucan synthesis and accumulation, and its regulatory mechanism needs to be further investigated. Most of the functions of members of the Csl gene family are related to polysaccharide synthesis in plants (Liepman & Cavalier, 2012). The CslF and CslH gene families have been well demonstrated to be required in β-glucan biosynthesis (Doblin et al., 2009; Schreiber et al., 2014). CslF9 is the putative (1,3;1,4)- β-glucan synthase. Garcia-Gimenez et al. (2020) knocked out the CslF9 gene in barley, but the mutant showed no significant change in β-glucan content compared to wild-type barley. The expression of the CslF9 gene screened in this study was low and without a noteworthy pattern. Therefore, we concluded that CslF9 had little effect on the β-glucan content of oat seeds. With the accumulation of β-glucan, the relative expression of TRINITY_DN16295, encoding CslF2, also gradually increased. The relative expression of TRINITY_DN34723, TRINITY_DN15420, and TRINITY_DN34252, encoding glucan endonucleases, always decreased. Therefore, these four genes may be the most significant genes for β-glucan synthesis in oat seeds. In fact, the functions of genes other than CslF6 have not yet been accurately verified. However, in the present study, no significant up-regulation of CslF6 expression was observed.

TFs are often considered to be key factors in the regulation of gene expression, but due to less research, few TFs involved in regulating the expression of seed β-glucan-related genes have been identified. Currently, only MYB has been shown to be involved in regulating the biosynthesis of secondary cell wall polysaccharides (Ko et al., 2014). The expression of CslF6 is increased in barley and rice mutants overexpressing MYB61. Expression of CslF6 is increased in barley and rice mutants overexpressing the MYB61 gene, and leaf (1,3;1,4)-β-glucan content is reduced by 31% in knockout MYB61 a knockout rice mutants (Burton & Fincher, 2014; Vega-Sánchez et al., 2012; Zhao et al., 2019). In the present study, we screened for only one up-regulated gene encoding MYB, but not for the gene encoding CslF6. Therefore, we suggest that the expression and function of β-glucan synthesis and regulatory genes may have large interspecies differences or be susceptible to interference by the external environment. Notably, in this study, we observed a gene encoding NAC, which was up-regulated in both varieties and expressed much higher in the high β-glucan variety BY than DY. Liang et al. (2021) overexpressed a gene of the NAC family, PeNAC1, in oats, which resulted in significant increases in the levels of (1,3;1,4)- β-d glucan content in transgenic lines. (1,3;1,4)- β-d glucan in the transgenic lines, as well as (1,3;1,4)- β-d glucan biosynthesis genes, AsCslF3, AsCslF6, and AsCslF9 in the transgenic lines, were significantly increased in response to salt stress. However, it is worthwhile to investigate whether this increase in gene expression is due to the regulatory effect of NAC or the result of salt stress. In addition, we note that 26% of the up-regulated genes we screened were genes encoding members of the Dof family, but no evidence has been found regarding the regulation of β-glucan synthesis by these TFs.

Conclusions

In this study, we measured the β-glucan content in oat seeds at different irrigating periods and determined that the critical period for its development in oat seeds occurs on days six to 16 of the irrigating period. On this basis, we sequenced the transcriptome of oat seeds at different irrigating periods. KEGG pathway enrichment analyses showed that the DEGs of the different comparative groups were significantly enriched in protein processing in the endoplasmic reticulum, pentose phosphate pathway, flavonoid biosynthesis, starch and sucrose metabolism, pentose and glucuronate metabolism, pentose and glucuronate pathway, galactose metabolism, and protein processing in oat seeds during filling. In the analysis of gene expression patterns in the starch and sucrose metabolism pathway. We screened three genes encoding dextran endonucleases, TRINITY_DN34723, TRINITY_DN15420, and TRINITY_DN34252, and retrieved one CslF2, two CslF8, and one CslF9 gene from DEG analysis. By analyzing the expression patterns of the transcription factors, we concluded that the B3, Dof, FAR1, and NAC families may play an active role in β-glucan synthesis and accumulation. We also screened 32 TFs belonging to the above families. Through weighted gene co-expression network analysis of DEGs, we obtained five modules, which were significantly and positively correlated with β-glucan content in oat seeds and contained 6,559, 1,223, 364, 80, and 139 DEGs, respectively. Finally, we performed RT-qPCR on seven genes, and the results showed that the transcriptome data were authentic and reliable. In future studies, we will continue to verify the functions of the above genes.

Supplemental Information

Supplemental Information 1 GO-terms dot diagram with significant enrichment of DEGs in 5 filling stages among different varieties

The color of the dot indicates PValue and the size of the dot indicates the number of differential genes enriched to that term.

Supplemental Information 2 GO-terms dot diagram with significant enrichment of DEGs during BY and DY grain development

The color of the dot indicates PValue and the size of the dot indicates the number of differential genes enriched to that term.

Supplemental Information 3 Venn diagram of 213 up-regulated expressed TFs between the two varieties at the same time period

Supplemental Information 4 Optimal Soft Threshold

Supplemental Information 5 β-glucan content of 130 oat germplasm resources

Supplemental Information 6 RNA Quality Assurance

Supplemental Information 7 Gene and primer selection for qRT-PCR

Supplemental Information 8 Oat grain β-glucan content and accumulation rate

Supplemental Information 9 Summary of DEGs of different varieties in different periods

Supplemental Information 10 Summary of up-regulated and down-regulated genes among varieties in different periods

Supplemental Information 11 Summary of DEGs in BY and DY at different grain filling stages

Supplemental Information 12 Summary of up-regulated and down-regulated genes in different filling stages

Supplemental Information 13 213 DEGs encoding TF were up-regulated

Supplemental Information 14 Raw data for Figure 8

Supplemental Information 15 MIQE

Additional Information and Declarations

Competing Interests

Author Contributions

Data Availability

The authors declare there are no competing interests.

Bing jie Qi conceived and designed the experiments, analyzed the data, prepared figures and/or tables, authored or reviewed drafts of the article, and approved the final draft.

Ming xue Ji conceived and designed the experiments, performed the experiments, analyzed the data, prepared figures and/or tables, authored or reviewed drafts of the article, and approved the final draft.

Zhu qing He conceived and designed the experiments, performed the experiments, analyzed the data, prepared figures and/or tables, authored or reviewed drafts of the article, and approved the final draft.

The following information was supplied regarding data availability:

The raw sequence reads are available at NCBI: PRJNA1011233.

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
