# Peer review of "Using transcriptome sequencing (RNA-Seq) to screen genes involved in β-glucan biosynthesis and accumulation during oat seed development"

_PeerJ, doi:10.7717/peerj.17804_

## Round 0.1 · original submission · Major Revisions

Dear authors,

Your manuscript is largely based on Geng et al. (2022) was inspired by their study. While their plant choices were barley, you chose oats, but it didn't seem nice that you used the same title. It would be more appropriate to slightly change the title of your manuscript. The reviewers have quite accurately highlighted other deficiencies in your manuscript. Just as it is necessary to improve the language of your manuscript, you should also make it comply with the rules of scientific article writing. In many places, there are sentences that do not match the scientific article writing style. For example, in line 350-353 it says "Geng found that barley varieties with up-regulated expression of photosynthesis-related genes also had elevated β-glucan content, thus concluding that photosynthesis was positively regulated in relation to β-glucan accumulation in barley seeds ( Geng et al. 2022).” is wrong. More precisely, "Geng et al. (2022) found that barley varieties with up-regulated expression of photosynthesis-related genes also had elevated β-glucan content, thus concluding that photosynthesis was positively regulated in relation to β-glucan accumulation in barley seeds" should be. Some similar ones have been mentioned by reviewers. Please review your manuscript carefully from beginning to end and revise it as appropriate.

Sincerely

·

Basic reporting

English is poor. This manuscript needs editing and proofreading by native speakers.
What is the FDA? I am sure that you know what it means. Me too, but you should give the long name in parenthesis for the first use.
Is it possible to use an updated citation instead of this “Hallfrisch & Behall 2000”? For such serious information supported by the FDA department, you need to confirm by new reports or new references.
For example, for instance, and so on... those usages not appreciated so much in academic writing. Please try to avoid...
There is no aim of this study. Please indicate it clearly at the end of the introduction section.
Line 69-76. This section belongs to the method, not the introduction. “In this study, ³-glucan content in oat grains at different 70 filling stages was detected, and two oat cultivars with significantly different ³-glucan content 71 were selected as experimental materials for transcriptome sequencing and data analysis. Based 72 on this study, we systematically analyzed the metabolic pathways of ³-glucan biosynthesis and 73 accumulation and mined many differentially expressed genes related to ³-glucan biosynthesis. 74 This study provided a theoretical basis for the analysis of the molecular mechanism of ³-glucan 75 biosynthesis and accumulation in oat grains and the cultivation of new varieties of oat with high 76 quality”
Please try to use more updated citations Holland et al. 1997, Hughes et al. 1997... and so on. Science is changing rapidly and those old studies can not be shedding light on agricultural production of today.
I think that this sentence is irrelevant “The cellulose-like synthase (CSL) gene family is thought to 57 encode ³-glucan synthase (Burton et al. 2006)”. Maybe this can be used at the beginning of the passage by a different statement.
But ı can say that the introduction section is written very well.

Experimental design

What is the pattern of the trial?
Put space “filling(control).
How much does the leaves sample weigh for RNA isolation?
Excel 2020, IBM SPSS Statistics 25, Primer Premier 5.0 software. Please give a reference for each software.
What are these? All DEGs are mapped to GO and 124 KEGG pathway databases
The statistical analysis section is absent.

Validity of the findings

For Figure 2, information about PCA, correlations is required.
Where is the information on the gene, such as amplicon size, and forward-reverse sequence at the end of primer design software?
The conclusion section is very poor.
Figures need references, you must indicate which software you used.
You need to give information in the method section about heat map and so on...
Which software did you use to perform gene expression analysis?

Results and Discussion is well structured.

Additional comments

What kind of advantages will the Beta-glucan synthesis genes and the accumulation of genes in the development stage of oats provide you?

For what purpose was such a study conducted? The work is very good. A lot of effort has been put in, but what exactly is the purpose here? Normally ı can say that this study was not so necessary, because only you want to learn gen gen expression during growing oat without any stress, or different applications. I don't understand what kind benefit will occur at the end of this study. But because of your effort so much, did so much analysis, and well-structured manuscript, ı can give you a Major revision.

Reviewer 2 ·

Basic reporting

* The manuscript needs major linguistic adjustments. In addition, the sentences should be written shorter and clearer by avoiding the use of unnecessary words.
• Lines 19-22: The sentence is badly written in standard English; accordingly, kindly reformulate it.
• Lines 166-170: The sentence is cumbersome; accordingly, kindly reformulate in order to make it clearer and more aiming.
• Line 177: “……..GC content The GC content…..” where dos the first sentence ends and where does the second starts?
• Lines 276-280: The sentence is cumbersome; accordingly, kindly reformulate in order to make it clearer and more aiming.
• Line 292: “.. module (r=0.93, p=9e-05)” what does p=9e-05 stands for?
• Lines 323-330: The sentence is cumbersome; accordingly, kindly reformulate in order to make it clearer and more aiming.
• Lines 337-341: The sentence is badly written in standard English; accordingly, kindly reformulate it.
• Line 349, 352, 365: Please write the references in the sentences according to the Journals instructions.
*Care should be taken in the use of abbreviations:
• Line 24: “Differentially expressed genes (DEGs)” not Differential expression genes…
• Line 120: The explanations of the abbreviations must be stated in the first place: “Fragments per kilobase million (FPKM)”
• There are so many abbreviations without explanations.
*Lines 15-34: The “Abstract” should be written as one section according to the journals instructions.
* Lines 31-33: “In total,………oat seeds”: Since the results given in the abstract part is insufficient, the conclusion made in this sentence seem to have no ground.
* Lines 36-37: A more recent literature must be cited or a recent data from FAO must be given for production value and ranking. Six to nine years past since the cited references.
*Lines 69-73, “In this study,…… biosynthesis”: The aim of the study must be putted on the line rather than mentioning what is done.
*Lines 74-76, “This study …… quality”: This sentence is more suitable for conclusion part.

Experimental design

*Lines 82-83: It is better to give the geographical location rather than the name of the town and/or district.
*Lines 89-93: Avoid using simple present tense in the manuscript. “ Sampling begins….., which is…. And takes sample……. Freeze, ….take……. “etc.
* Lines 89-90: Does the first day of filling is taken as control or 1d. Because in lines 217-218 it is understood as 1d is taken as control. This must be clarified. Also The sentence is badly written in standard English; accordingly, kindly reformulate it.
*Line 94: The authors mentioned in this line that they took three biological replicates and in line 101 and 155 they mention four replicates. Four technical replicates from one biological replicates or how did they choose this additional one replicate.
* Line 137: What were the concentrations of the RNAs, at least write a range.
*Line 138: “RNA with A260/280 ratio……. could be used for…..experiments”. This is a “must” for genetic analysis. But what about your RNAs purities.
*Line 139: “The barley beta-Actin….used as internal reference……..” WHY? While there is actin sequence in NCBI for Avena sativa, Avena nuda clones and Avena fatua.Please explain it or give the reference work.

Validity of the findings

*Line 196, 197,198 and 225: “up-regulated expressed genes” or “down-regulated expressed genes” this is not an appropriate use to indicate the relative expression. Either use “up- and/or down- regulated genes” or “increased and/ or reduced/decreased relative gene expression”
*Line 211-213, Lines 232-234, Lines 240-242, Lines 253-257, Lines 264-266, Lines 283-284: The inferences made at the end of almost each Results subsection should be moved to the “Discussion” section.
*Line 258-259: It is mentioned as “It is well documented……….synthesis”, however not a single citation was made.
*Lines 275-276- Lines 283-284: All the TFs and genes mentioned as “important in β-glucan synthesis and7 or accumulation must be linked well to it.
*Lines 374-375: “and we will try………... of the genes” is a future prospects sentence, should be moved to “Conclusion” section.
*All the parameters examined should be discussed and supported by the literature in the Discussion section.
* A more comprehensive and detailed “Conclusion” section should be written.

Additional comments

I commend the authors for their extensive data set and complied laboratory work. However, there is a weakness in emphasizing and highlighting the important and strengths of the work. To provide more justification of your study discussions must be fluently linked between introduction and results and the conclusions must be drawn appropriately based on the data presented.

Reviewer 3 ·

Basic reporting

When the sources were examined, it was seen that there was very little current literature on the subject. The article needs to be enriched with new literature.

Experimental design

The experimental design has been successfully constructed.

Validity of the findings

The research is statistically comprehensive.

---

## Round 0.2 · Minor Revisions

Your manuscript needs one more minor correction before it can be accepted. Please revise your aim of the study to be more appropriate.

·

Basic reporting

The author did the entire revisons. From my side, it is ok.

Experimental design

-

Validity of the findings

-

Reviewer 2 ·

Basic reporting

*Each comment from the referees was addressed one at a time, and the required changes were put in place.
*Care was taken in the writing of references in the text and the reference list.
*Even though the authors changed the last sentence in the introduction in line with the criticisms, it still does not qualify as a purpose/aim statement. This sentence should be in the colclusion section, not in the introduction section and a new sentence indicating the purpose of this study should be written.

Experimental design

*Lack of explanation in the method sections have been completed.
*The qRT-PCR analysis was re-performed with oat actin gene instead of barley actin gene as internal reference.

Validity of the findings

*Discussion section is detailed
*Attention was drawn to the importance of the study

Reviewer 3 ·

Basic reporting

The current literature requested in the revision has been added to the article.

Experimental design

No comment

Validity of the findings

No comment

---

## Round 0.3 · accepted · Accept

I see that you have completed all the changes for acceptance for publication. Congratulations